# The Impact of Nano- and Micro-Silica on the Setting Time and Microhardness of Conventional Glass–Ionomer Cements

**DOI:** 10.3390/dj12030054

**Published:** 2024-02-27

**Authors:** Zeynep A. Güçlü, Şaban Patat, Nichola J. Coleman

**Affiliations:** 1Department of Pediatric Dentistry, Faculty of Dentistry, Erciyes University, Melikgazi, Kayseri 38039, Türkiye; zaguclu@gmail.com; 2ERNAM, Erciyes University Nanotechnology Application & Research Center, Erciyes University, Melikgazi, Kayseri 38039, Türkiye; 3Department of Chemistry, Faculty of Science, Erciyes University, Melikgazi, Kayseri 38010, Türkiye; patat@erciyes.edu.tr; 4School of Science, Faculty of Engineering and Science, University of Greenwich, Chatham Maritime, Kent ME4 4TB, UK

**Keywords:** nano-silica, micro-silica, glass–ionomer cement, setting time, microhardness, scanning electron microscopy, nanoparticles

## Abstract

The objective of this study was to investigate the effect of the incorporation of 2, 4 or 6 wt% of amorphous nano- or micro-silica (Aerosil^®^ OX 50 or Aeroperl^®^ 300 Pharma (Evonik Operations GmbH, Essen, Germany), respectively) on the net setting time and microhardness of Ketac™ Molar (3M ESPE, St. Paul, MN, USA) and Fuji IX GP^®^ (GC Corporation, Tokyo, Japan) glass–ionomer cements (GICs) (viz. KM and FIX, respectively). Both silica particles were found to cause a non-linear, dose-dependent reduction in setting time that was within the clinically acceptable limits specified in the relevant international standard (ISO 9917-1:2007). The microhardness of KM was statistically unaffected by blending with 2 or 4 wt% nano-silica at all times, whereas 6 wt% addition decreased and increased the surface hardness at 1 and 21 days, respectively. The incorporation of 4 or 6 wt% nano-silica significantly improved the microhardness of FIX at 1, 14 and 21 days, with no change in this property noted for 2 wt% addition. Micro-silica also tended to enhance the microhardness of FIX, at all concentrations and times, to an extent that became statistically significant for all dosages at 21 days. Conversely, 4 and 6 wt% additions of micro-silica markedly decreased the initial 1-day microhardness of KM, and the 21-day sample blended at 4 wt% was the only specimen that demonstrated a significant increase in this property. Scanning electron microscopy indicated that the nano- and micro-silica particles were well distributed throughout the composite structures of both GICs with no evidence of aggregation or zoning. The specific mechanisms of the interaction of inorganic nanoparticles with the constituents of GICs require further understanding, and a lack of international standardization of the determination of microhardness is problematic in this respect.

## 1. Introduction

Conventional glass–ionomer cements (GICs) continue to be a popular choice of dental restorative, particularly in paediatric dentistry and atraumatic restorative treatment (ART) [1,2]. They are formulated with a basic calcium or strontium fluoroaluminosilicate glass, a polyalkenoic acid (e.g., poly(acrylic acid)), water, and optional ratemodifiers, which are mixed to form a self-setting paste [1]. The setting reactions begin with the release of cations (e.g., Na^+^, Ca^2+^, Sr^2+^, Al^3+^) from the partial etching of the glass in the polyalkenoic acid solution. These dissolved cations then interact with the carboxylate groups of the acid causing ionic cross-linking between the polymer chains, which brings about the stiffening and setting of the mixture within a few minutes of mixing [1].

GICs are distinguished by good biocompatibility with oral tissues, release and recharge of fluoride ions, and the formation of chemical bonds with enamel and dentine [1]. Their less advantageous properties include an initial sensitivity to moisture, poor mechanical strength, inadequate wear-resistance, and insufficient hardness to withstand local deformation from occlusal masticatory forces [1].

During the past decade, a range of inorganic nanoparticles (INPs) (i.e., Ag, AgVO_3_, Al_2_O_3_, BaSO_4_, Ca_5_(PO_4_)_3_OH, Cu, CuO, MgO, Mg_2_SiO_4_, SiO_2_, TiO_2_, YbF_3_, ZnO ZrO_2_, bioactive glasses, amorphous calcium phosphate, graphene oxide, carbon nanotubes, and aluminosilicate nanoclays) has been incorporated into glass–ionomer cements with the objective of improving their mechanical, chemical, and/or biological properties [3,4,5,6,7,8,9,10,11,12,13,14,15,16,17,18,19,20,21,22,23,24,25]. In brief, enhanced antimicrobial properties are reported for Ag-, AgVO_3_-, Cu-, CuO-, MgO-, TiO_2_-, and ZnO-blended GICs [3,4,6,7,10,11,12,13,14,16]. The incorporation of silica (SiO_2_), hydroxyapatites (Ca_5_(PO_4_)_3_OH), and bioactive glasses is observed to give rise to superior cytocompatibility and bioactivity [18,20,21,22,23], and improved mechanical properties are noted for GICs blended with a wide range of metal and metal oxide nanostructures [3,4,5,6,8,9,12,15,17,19]. However, no universal consensus has been established on the impact of INPs on GICs, owing to significant variations in the properties of INPs used in the various studies, differences in GIC formulations, and the absence of any systematic large-scale laboratory-based investigations or clinical trials. Recent developments in nanoparticle-blended GICs are reviewed in more detail in a number of publications [3,4,24,25].

Despite current interest in the incorporation of INPs in dental restoratives, there are relatively few reports of the impact of pure silica particles on the properties of GICs [18,19]. Zhao and Xie [19] found that the incorporation of 1–3% nano- or micro-silica significantly increased resistance to attrition, and modestly reduced resistance to abrasion of a commercial GIC (Fuji II^®^ (GC Corporation, Tokyo, Japan)). The silica-blended samples also exhibited superior Knoop hardness. Both wear and hardness data were reported at 24 h only, with no indication of the longer-term effects of nano- and micro-silica on these properties [19]. Crystalline and amorphous nano-silica particles have also been reported to enhance the in vitro bioactivity (i.e., the ability to bond with living tissue) of a glass–ionomer luting cement (Medicem, Promedica Dental Material GmbH, Neumuenster, Germany) that is tentatively postulated to reduce marginal gap formation [18].

A wider body of literature exists on the impact of pure silica particles on other dental restoratives, although knowledge of the effect of pure silica particles in conventional GICs is currently limited [3,4,18,19,24]. To address this deficiency, the present study concerns the impact of 2, 4 and 6 wt% additions of pure amorphous nano- or micro-silica on the setting times and on the 1-, 7-, 14- and 21-day Vickers microhardness of two conventional GICs. The effect of nano- or micro-silica on the microstructures of the GICs is also considered by scanning electron microscopy (SEM) with energy dispersive X-ray analysis (EDX).

The two well-characterized, differently formulated restoratives selected for this study, Ketac™ Molar (3M ESPE, St. Paul, MN, USA) [26] and Fuji IX GP^®^ (GC Corporation, Tokyo, Japan) [27], are both conventional GICs that are principally used in non-load-bearing applications, root restorations, core build-up, luting and ART.

The nano-silica selected for this study, Aerosil^®^ OX 50 (Evonik Operations GmbH, Essen, Germany), comprises non-porous spherical particles of hydrophilic fumed silica (~40 nm in diameter) that finds commercial application in dental composites [28]. This material was chosen on the basis of its high purity, poor thickening properties, low tendency to form agglomerates and acidic pH, as these characteristics minimise chemical interference in the intrinsic setting reactions of the GICs.

The colloidal micro-silica used in this study, Aeroperl^®^ 300 Pharma (Evonik Operations GmbH, Essen, Germany), comprises highly pure, acidic, porous particles (20–60 μm in diameter) characterized by easy handling, excellent flow properties, and low aggregation [29]. It is a common constituent of foodstuffs, cosmetics, and pharmaceutical formulations.

## 2. Materials and Methods

### 2.1. Materials

The constituents of the commercial GICs used in this study are listed in Table 1, and the properties of the commercial silica particles are summarized in Table 2.

### 2.2. Setting Time

The setting times of Ketac™ Molar (3M ESPE, St. Paul, MN, USA) and Fuji IX GP^®^ (GC Corporation, Tokyo, Japan) (viz. KM and FIX, respectively) were determined, in triplicate, in accordance with the method described in ISO 9917-1:2007 [30]. Both commercial GICs are presented as a loose powder and a solution that are mixed by hand at the mass ratios indicated in Table 1. Accordingly, each GIC was mixed by manual spatulation on a ceramic tile for 30 s, in accordance with the manufacturers’ instructions, and placed in a stainless-steel mould (200 mm × 200 mm × 8 mm) that was mounted on an aluminium-foil-wrapped steel block heated to 37 °C at a relative humidity of ~100% [26,27,30]. The time taken for a 400 g Gilmore needle (UTEST-Material Test Equipment Ltd., Ankara, Türkiye) with a flat-end indenter (1 mm in diameter) to fail to make a complete circular impression on the surface of the sample was recorded as the setting time.

The nano- or micro-silica particles were initially combined with the powder component of each GIC using a vortex mixer, followed by manual spatulation on a ceramic tile (as indicated above), prior to the introduction of the solution component of the GIC. The silica particles were incorporated at 2, 4 and 6 wt% with respect the mass of GIC powder, and each analysis was carried out in triplicate (following two initial preliminary trials for each sample). The specific quantities of components for each sample are listed in Table 3.

The two-tailed Student *t*-test was used to compare the setting times of the silica-blended samples with those of the unblended controls at *p* = 0.05. Data were expressed as mean values, and in each case, the standard deviation was within 2% of the mean. 

### 2.3. Microhardness

The mix proportions of the samples prepared for microhardness testing are listed in Table 3. Three cylinders (4 mm diameter × 6 mm height) of each sample-type were prepared in split stainless-steel moulds capped at the top and bottom with steel plates and cured for 1 h at 37 °C and 100% RH. They were then demoulded, individually immersed in 5 cm^3^ of deionised water in an air-tight polypropylene tube, and stored at 37 °C until required. There is currently no standard procedure for the determination of the microhardness of GICs, so the sample preparation and curing regime were adopted from those described in ISO 9917-1:2007 [30] for the determination of compressive strength. This preparation method produces appropriately cured samples with two parallel, flat, smooth surfaces that are suitable for microhardness testing.

Microhardness was determined at 1, 7, 14, and 21 days on triplicate samples of each type. In each case, the sample was removed from its storage solution, patted dry on lint-free tissue, and subjected to Vickers hardness testing (DuraScan G5 (EMCO-TEST Prüfmaschinen GmbH, Kuchl, Austria) by making three indentations on the top and bottom of each sample using a 300 g mass for 15 s. Representative example images of the measurement of the indentation for the evaluation of microhardness are given in Appendix A. The microhardness data were subjected to analysis of variance (ANOVA) at *p* = 0.05, and statistically significant differences were tested by the post hoc Tukey’s honestly significant differences test at *p* = 0.05.

### 2.4. SEM and EDX Analysis

Fracture surfaces of the control GIC samples and those blended with 6 wt% nano- or micro-silica were created at 21 days by crushing with a loading rate of 50 N min^−1^ and a cross-head speed of 0.75 mm min^−1^ (AG-XD 50 tensile tester, Shimadzu, Istanbul, Türkiye). The samples were then attached to aluminium stubs with conductive carbon tabs (Leit-C™, SPI Supplies Division, PA, USA). Backscattered electron images and energy dispersive X-ray (EDX) maps were collected from the uncoated samples with a field emission gun scanning electron microscope (Zeiss Sigma 300, Cambridge, UK) fitted with an EDX detector (X-Max 50 detector, Oxford Instruments, Oxford, UK). Samples were analysed in low vacuum mode with an accelerating voltage of 20 kV at nominal magnifications of ×200 and ×1k.

## 3. Results

### 3.1. Setting Time

The impact of 2, 4 and 6 wt% additions of nano- or micro-silica on the setting time of KM is shown in Figure 1. These data indicate that both nano- and micro-silica caused a non-linear dose-dependent decrease in setting time. Similar trends were also observed for the effect of 2, 4 and 6 wt% additions of nano- or micro-silica on the setting time of FIX (Figure 2). In general, micro-silica had a more pronounced impact on the reduction in setting time than nano-silica. In all cases, the setting time of the silica-blended GIC sample was found to be significantly lower than that of the corresponding unblended control (*p* = 0.05).

### 3.2. Microhardness

The microhardness of KM incorporating 0, 2, 4 and 6 wt% additions of nano- or micro-silica is presented as functions of time in Figure 3 and Figure 4, respectively. In the absence of additional silica, the microhardness of KM was observed to increase steadily from 67.3 ± 4.7 to 85.1 ± 9.5 between 1 and 21 days as the maturation processes within the cement continued (Figure 3 and Figure 4). 

At 1 day, all concentrations of nano-silica were observed to have reduced the microhardness of KM, although this effect was only statistically significant (*p* = 0.05) at a dosage of 6 wt% (Figure 3). At 7 and 14 days, 4 and 6 wt% concentrations of nano-silica increased microhardness, and 2 wt% additions diminished this property, although these effects were all statistically insignificant relative to the respective unblended KM control samples at these times. By 21 days, the microhardness of KM blended with 6 wt% nano-silica was found to be significantly higher than those of the other 21-day samples, which were all statistically similar (Figure 3).

At 1 day, 4 and 6 wt% concentrations of micro-silica significantly reduced the microhardness of KM and, conversely, a 2 wt% addition was found to cause a statistically irrelevant increase in this property (Figure 4). Marginal increases in microhardness were observed at 7 and 14 days for KM blended with 4 and 6 wt% micro-silica, and modest decreases were noted for samples incorporating 2 wt%, although none of these differences was statistically significant with respect to the unblended control samples. By 21 days, all micro-silica-blended KM samples were harder than their unblended counterpart, although only the sample incorporating a 4 wt% addition was statistically higher than the control (Figure 4).

The microhardness of FIX incorporating 0, 2, 4 and 6 wt% additions of nano- or micro-silica, at 1, 7, 14 and 21 days, is shown in Figure 5 and Figure 6, respectively. The microhardness of FIX increased from 75.0 ± 5.8 to 92.2 ± 7.6 between 1 and 7 days, with no further statistically significant changes in this property throughout the duration of the investigation (Figure 5 and Figure 6).

At 1 and 14 days, insignificant decreases in microhardness were observed for FIX blended with 2 wt% nano-silica; however, at these time-points, samples containing 4 and 6 wt% nano-silica were found to be statistically harder than their corresponding unblended controls (Figure 5). At 7 days, no statistically relevant differences in microhardness were found among all FIX samples, irrespective of composition, although, by 21 days, enhanced microhardness was observed for samples incorporating 4 and 6 wt% nano-silica (Figure 5).

At 1 day, dose-dependent increases in microhardness were observed for all FIX samples blended with micro-silica particles, although this enhancement was only significant at the highest concentration of 6 wt% (Figure 6). At 7 days, modest increases in microhardness were noted for the samples blended with 4 and 6 wt% micro-silica relative to the control, although none of the differences observed among the 7-day sample group was significant. Incorporation of 4 and 6 wt% microparticles significantly improved the microhardness of FIX at 14 days, and all micro-silica-blended cements were statistically harder than the unblended control at 21 days (Figure 6).

### 3.3. SEM and EDX Analysis

Backscattered electron (BSE) images, at ×200 magnification, of KM and FIX incorporating 0 and 6 wt% additions of nano- or micro-silica are shown in Figure 7. The BSE image of KM displays a uniform distribution of 5–10 μm glass fragments embedded within the cross-linked polyacid matrix throughout which is distributed a number of 5–10 μm air voids that were entrained during mixing (Figure 7). The BSE image of FIX exhibits similar features, although with a broader glass particle size range of 5–50 μm (Figure 7). The images of both KM and FIX closely resemble those reported in the literature for these materials [31,32]. 

The incorporation of 6 wt% nano-silica is seen to have no apparent impact on the microstructure of either KM or FIX (Figure 7). In particular, there is no evidence of the aggregation, agglomeration or zoning of the silica nanoparticles (that are, individually, too small to be detected by SEM) within the GICs. Conversely, micro-silica particles (indicated by arrows in Figure 7) are observed to be uniformly distributed throughout the cement matrices of KM-6MS and FIX-6MS, and their incorporation in both formulations appears to have reduced the number of entrained air voids.

BSE images, at ×1k magnification, and corresponding Si, Al and Ca EDX maps of KM incorporating 0 and 6 wt% nano- or micro-silica are presented Figure 8, Figure 9 and Figure 10. The accompanying EDX spectra are located in Appendix A. The higher-magnification BSE image of KM (Figure 8) provides a more detailed perspective of the distribution of glass particles within the polyacid matrix than that given in Figure 7, and the EDX data confirm that the principal elemental components of the glass are oxygen, silicon, fluorine, aluminium, calcium, lanthanum, phosphorus and sodium (as reported by other researchers [33]).

As in Figure 7, Figure 9 shows no indication of aggregation or zoning of the nano-silica particles within the KM-6NS matrix and also no evidence that the incorporation of these nanoparticles has an impact on the microstructure of the cement.

Micro-silica particles (indicated by arrows) are visible within the matrix of KM-6MS (Figure 10), although EDX mapping does not clearly discriminate between these pure silica particles and the multi-component matrix owing to the scale and intimate association of all of the phases present.

BSE images, at ×1k magnification, and accompanying Si, Al and Sr EDX maps of FIX incorporating 0 and 6 wt% nano- or micro-silica are shown Figure 11, Figure 12 and Figure 13. The corresponding EDX spectra are presented in Appendix A. The EDX data confirm that the major elemental components of the glass are oxygen, silicon, fluorine, aluminium, strontium, phosphorus, sodium and titanium (as previously reported [33]).

The comparatively broad range of glass particle sizes in the FIX samples is again evident in Figure 11, Figure 12 and Figure 13. As with KM, there is no evidence of the aggregation or zoning of the nano-silica particles, and their incorporation into the cement is observed to have had no significant impact on the microstructure (Figure 12).

The micro-silica particles in sample FIX-6MS are denoted by arrows in Figure 13. Despite these being composed of pure silica, the corresponding EDX maps fail to make this distinction owing to the proximity of the other phases within the cement (Figure 13).

## 4. Discussion

As previously mentioned, recent studies have demonstrated that various INPs can be incorporated into conventional GICs to enhance their properties, and that there are comparatively few reports of pure silica particles in this application [3,4,5,6,7,8,9,10,11,12,13,14,15,16,17,18,19,20,21,22,23,24,25]. The findings of the present study have indicated that nano- and micro-silica can be uniformly distributed throughout the GIC matrix by simple manual spatulation. The ease of distribution of silica particles is likely to be facilitated by the presence of poly(acrylic acid-co-maleic acid) and poly(acrylic acid) in the KM and FIX formulations, respectively, as these polymers are widely exploited industrial dispersants for metal oxides in aqueous systems [34,35]. Nano-silica was found to have no significant impact on the microstructure of the KM and FIX cements, although blending with micro-silica significantly reduced the number of air voids that were entrained within the matrix during mixing (Figure 7). In fact, micro-silica particles are used as fillers to reduce air-entrainment in viscous fluids for various engineering applications [36]. A reduction in entrained air voids has also been reported for the incorporation of Al_2_O_3_, ZrO_2_, and TiO_2_ particles in commercial GICs [8,37]. Air voids act as stress-concentrators and thus points of mechanical weakness within the GIC matrix, so their potential reduction is advantageous.

ISO 9917-1:2007 [30] stipulates that the net setting time of a glass polyalkenoate cement, for use as a base, lining or restoration, must fall within the range of 90 to 360 s. This standardized range ensures that the GIC possesses sufficient working time for the clinician to mix, manipulate and place the cement, and that the cement is able to set within a clinically appropriate time for the consultation. Inappropriately long setting times also potentially expose the immature cement to moisture ingress from the oral cavity that could compromise the setting reactions, subsequent chemical and mechanical properties and, ultimately, the lifespan of the restoration.

The findings of the present study indicate that the addition of 2, 4 or 6 wt% nano- or micro-silica caused statistically significant dose-dependent decreases in the setting times of both commercial GIC cements that still fall within the clinically acceptable range (Figure 1 and Figure 2) [30]. The authors failed to find data in the current scientific literature on the impact of pure nano- or micro-silica on the setting times of conventional GICs, so a comparison with existing findings is not possible at the present time. However, the setting times of unblended KM (313 ± 5.8 s) and FIX (280 ± 2.9 s) compare favourably with those reported in other studies [33,38,39].

Some INPs are reported to prolong setting time (e.g., Ag, MgO) [40,41], some are indicated to have no statistically significant effect (e.g., TiO_2_, ZnO) [38,42], and others are observed to reduce it (e.g., BaSO_4_, YbF_3_, TiO_2_) [9,43]. In some cases, conflicting findings are reported for the impact of INPs on setting time, such as TiO_2_, that is reported to both reduce [43] and have no impact on this property [38]. It is evident that these observed discrepancies arise from variations in the surface properties, chemistry and reactivity of the different INPs, and from differences among the commercial GIC formulations, although the interrelationships between these parameters are not currently understood.

It could be conjectured that basic INPs are able to react with the polyalkenoic acid, thus limiting the extent of acid-etching of the GIC’s glass phase, and therein reducing or retarding the release of the cross-linking di- and trivalent cations that are essential to the setting reactions [1]. The presence of the INPs could also limit the diffusion and transport mechanisms within the GIC network, thus reducing the efficiency of setting. Conversely, it is also possible that the INPs could interact with the rate-modifying additives that slow the setting reactions by chelating the di- and trivalent cations; in which case, the incorporation of INPs could reduce the setting time of the GIC.

It is possible that the reduction in setting times observed in the present study derives from the effect of the surface charge of silica within the GIC systems. Silica has a low ‘point of zero charge’, which means that it presents a negative surface charge at pH values above ~2.5 [44,45]. The initial pH of GICs is reported to be in the range 0.9–2, which rises rapidly to 3.8–4.3 during mixing [46]. Under these conditions, the cations released from the basic glass component of the GIC are electrostatically attracted to the negative surface charge of the silica particles. Accordingly, the dissolved ions participate in an ‘electrical double layer’ at the negatively charged silica surface in which the cations are predominantly concentrated. The preferential interaction of cations with the surface of the silica particles reduces their concentration (i.e., their ionic activity) in the bulk of the solution, which provides a driving force for their further dissolution from the basic glass. Enhancing the rate of dissolution of cations from the basic glass will inevitably speed up the initial setting reaction that relies on a supply of di- and trivalent cations to cross-link the polyacid. The negative surface charge of silica is also likely to inhibit the aggregation of its particles in GICs as they experience mutual electrostatic repulsion.

A full understanding of the ways in which INPs influence the setting mechanisms of GICs is currently thwarted by limited analytical data and the lack of any large systematic studies. Nonetheless, the fact that the determination of setting times of conventional GICs is internationally standardized does facilitate the comparison of setting data among the various studies reported in the literature [30].

The determination of the microhardness of GICs is not standardized, and this, along with the various types of measurement (e.g., Vickers, Knoop, Rockwell, Brinell, Shore), presents a challenge to the comparison and critical analysis of microhardness data reported in the literature [47]. The lack of standardization also gives rise to considerable variation in the experimental parameters used to cast, cure, and store the GIC specimens for microhardness testing. This is further complicated by the application of coatings and/or finishing procedures that affect the surface of the GIC. In the field of dental materials, microhardness is typically, although not exclusively, measured by indentation (rather than scratch testing) using a Vickers or Knoop indenter [47].

In the present study, the sample preparation and curing regime were adopted from those described in ISO 9917-1:2007 [30] for the determination of compressive strength, and the Vickers method was used for the evaluation of microhardness. The microhardness of both commercial GICs increased with time, as the maturation process within the cements continued (Figure 3, Figure 4, Figure 5 and Figure 6). The microhardness of FIX reached a maximum value within 7 days, whereas KM continued to harden throughout the 21-day observation period (Figure 3, Figure 4, Figure 5 and Figure 6). These GICs are formulated differently (Table 1), so this finding is unsurprising.

The incorporation of nano-silica impacted differently on the microhardness of the two GICs (Figure 3 and Figure 5). The microhardness of KM was statistically unaffected by blending with 2 or 4 wt% nano-silica at all time-points, whereas 6 wt% addition decreased and increased the surface hardness at 1 and 21 days, respectively (Figure 3). In comparison, the incorporation of 4 or 6 wt% nano-silica significantly improved microhardness of FIX at 1, 14 and 21 days, with no change in this property noted for 2 wt% addition (Figure 5). No statistical differences were observed among the FIX groups at 7 days, after which time only the samples blended with 4 and 6 wt% nano-silica continued to gain appreciable microhardness relative to the unblended control.

Overall, these findings indicate that FIX is more sensitive to the incorporation of nano-silica than KM, and that the addition of these nanoparticles has a more positive effect on the microhardness of the former material (Figure 3 and Figure 5). Likewise, micro-silica was also found to have a greater impact on the microhardness of FIX than that of KM (Figure 4 and Figure 6). Micro-silica tended to enhance the microhardness of FIX at all concentrations throughout the timeframe of the experiment, to an extent that became statistically significant for all dosages at 21 days (Figure 6). Conversely, 4 and 6 wt% additions of micro-silica markedly decreased the initial 1-day microhardness of KM, and the 21-day sample blended at 4 wt% was the only specimen that demonstrated a significant increase in this property relative to that of the unblended control (Figure 4).

It is clear that the incorporation of nano- or micro-silica has different impacts on the microhardness of KM and FIX that relate to the concentration of the silica particles and also to the intrinsic properties of the GICs that arise from their different formulations. Furthermore, it should be noted that the selected curing and storage conditions may additionally influence the development of microhardness, especially since immature GICs are highly sensitive to moisture [1].

As previously mentioned, a study on the impact of 1–3% pure nano- or micro-silica on the microhardness of Fuji II^®^ (GC Corporation, Tokyo, Japan), a conventional GIC, was found to improve Knoop hardness values at 24 h. No other reports on the effect of pure nano- or micro-silica on the microhardness of conventional GICs could be found, for comparison, in the current scientific literature. However, other metal and metal oxide INPs (e.g., Ag, MgSiO_4_, TiO_2_, Ca_5_(PO_4_)_3_OH) are reported to enhance microhardness [15,48,49,50,51], whilst some (e.g., Ag, nanoclay) are indicated to have no effect [17,52], and others (e.g., BaSO_4_, YbF_3_, TiO_2_, ZnO, Ca_5_(PO_4_)_3_OH) are noted to diminish this property [9,42,43,48,53,54]. As with setting time, discrepancies exist between the reported impact on microhardness of certain INPs (e.g., Ag, Ca_5_(PO_4_)_3_OH, TiO_2_) that, in all probability, derive from the surface properties, reactivity and concentration of the INPs; the formulation of the GICs; and the highly variable, non-standardized experimental parameters that have been employed to determine this property.

Silica, one of the most abundant minerals in the Earth’s crust, is a popular constituent of many pharmaceutical formulations, foodstuffs, cosmetics, personal hygiene products and medical devices as it is simple and cheap to produce [44]. Furthermore, unlike many other metal oxide particles, silica exhibits low toxicity across a wide range of human cell-types and can stimulate hard tissue regeneration at a genetic level [55]. Hence, silica particles afford a number of clinical advantages over other INPs, and accordingly, further work is warranted to investigate the potential influence of nano- and micro-silica on the mechanical, chemical and biological properties of GICs. Since the current study is limited to setting time and microhardness, it is now suggested that further research be carried out on the impact of silica particles on other important properties such as flexural, compressive, shear and tensile strengths; adhesive and bonding characteristics; solubility; fluoride-release and recharge; and biocompatibility of GICs. 

## 5. Conclusions

The incorporation of 2, 4 or 6 wt% medical-grade amorphous nano- or micro-silica (Aerosil^®^ OX 50 or Aeroperl^®^ 300 Pharma (Evonik Operations GmbH, Essen, Germany), respectively) reduces the net setting times of Ketac™ Molar (3M ESPE, St. Paul, MN, USA) and Fuji IX GP^®^ (GC Corporation, Tokyo, Japan) glass–ionomer cements within the clinically acceptable limits specified in the relevant international standard (ISO 9917-1:2007 [30]). These silica particles also effect various dose- and time-dependent increases and decreases in surface microhardness. In addition, micro-silica was found to reduce the number of entrained air voids within the GICs. The specific mechanisms of the interaction of inorganic nanoparticles with the constituents of GICs require further understanding, and a lack of international standardization of the determination of microhardness is problematic in this respect.

## Figures and Tables

**Figure 1 dentistry-12-00054-f001:**
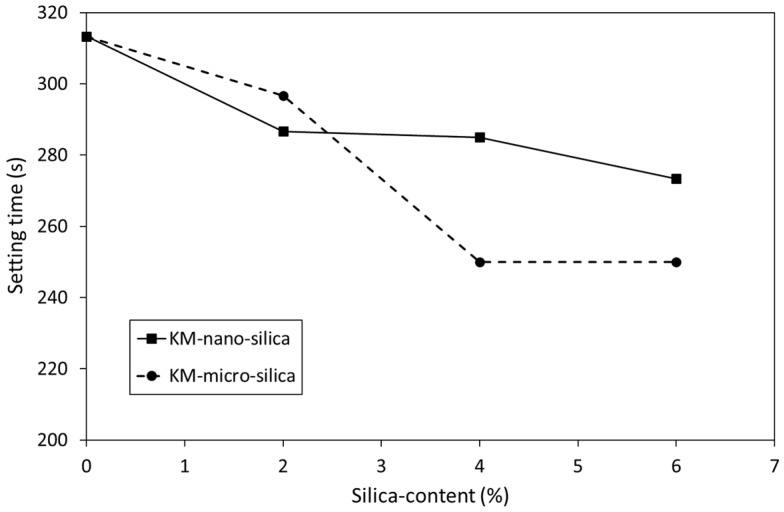
Setting times of Ketac™ Molar incorporating Aerosil^®^ OX 50 nano-silica (KM-nano-silica) and Aeroperl^®^ 300 Pharma micro-silica (KM-micro-silica).

**Figure 2 dentistry-12-00054-f002:**
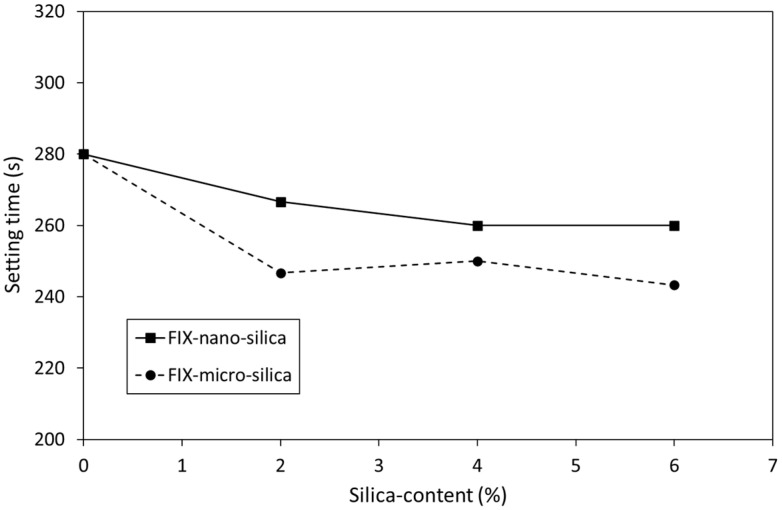
Setting times of Fuji IX GP^®^ incorporating Aerosil^®^ OX 50 nano-silica (FIX-nano-silica) and Aeroperl^®^ 300 Pharma micro-silica (FIX-micro-silica).

**Figure 3 dentistry-12-00054-f003:**
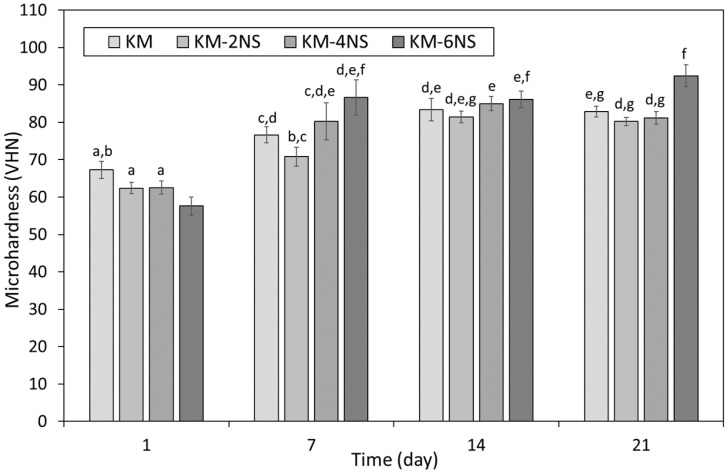
Microhardness of Ketac™ Molar incorporating 0, 2, 4 or 6 wt% Aerosil^®^ OX 50 nano-silica (viz. KM, KM-2NS, KM-4NS and KM-6NS). Statistically insignificant differences are denoted by the same letters.

**Figure 4 dentistry-12-00054-f004:**
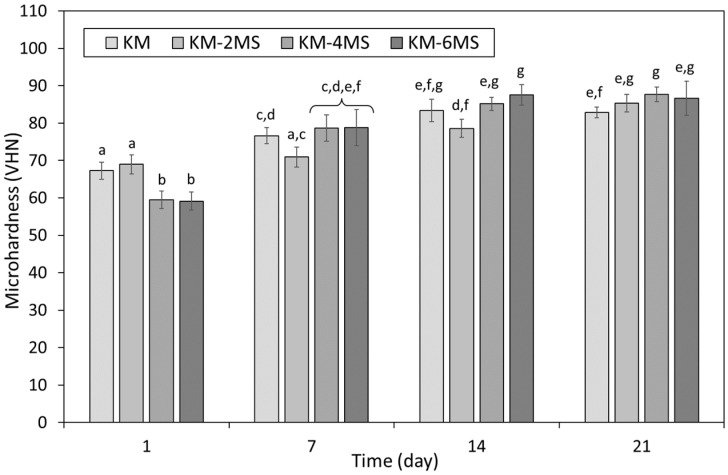
Microhardness of Ketac™ Molar incorporating 0, 2, 4 or 6 wt% Aeroperl^®^ 300 Pharma micro-silica (viz. KM, KM-2MS, KM-4MS and KM-6MS). Statistically insignificant differences are denoted by the same letters.

**Figure 5 dentistry-12-00054-f005:**
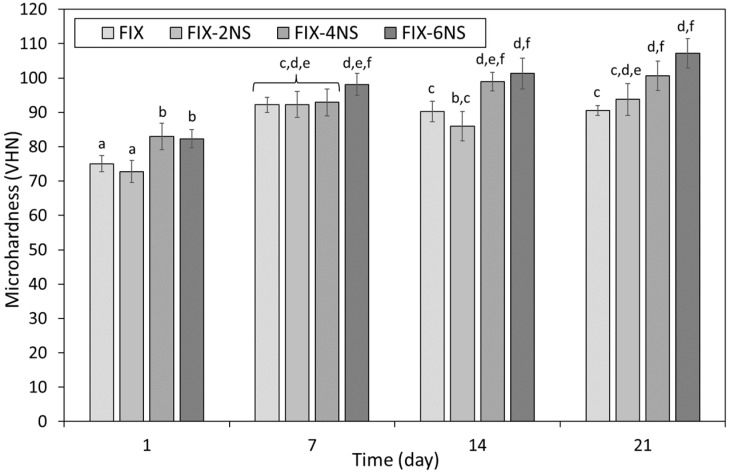
Microhardness of Fuji IX GP^®^ incorporating 0, 2, 4 or 6 wt% Aerosil^®^ OX 50 nano-silica (viz. FIX, FIX-2NS, FIX-4NS and FIX-6NS). Statistically insignificant differences are denoted by the same letters.

**Figure 6 dentistry-12-00054-f006:**
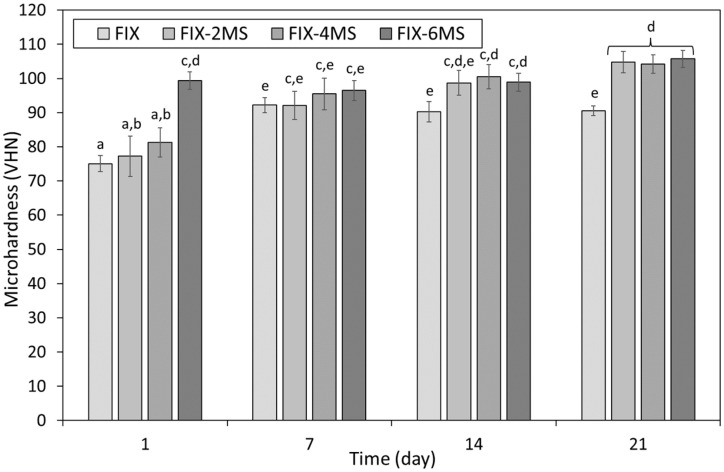
Microhardness of Fuji IX GP^®^ incorporating 0, 2, 4 or 6 wt% Aeroperl^®^ 300 Pharma micro-silica (viz. FIX, FIX-2MS, FIX-4MS and FIX-6MS). Statistically insignificant differences are denoted by the same letters.

**Figure 7 dentistry-12-00054-f007:**
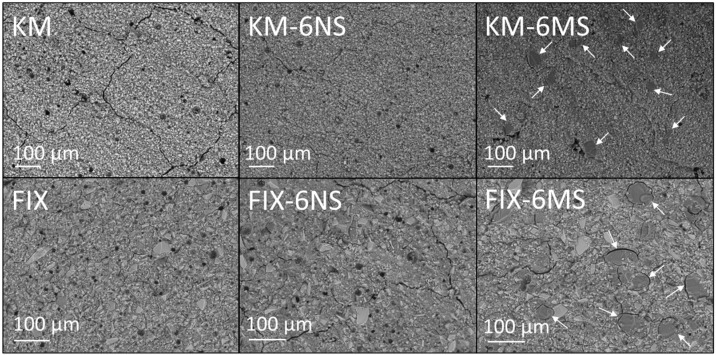
Backscattered electron images of Ketac™ Molar incorporating 0 and 6 wt% Aerosil^®^ OX 50 nano-silica or Aeroperl^®^ 300 Pharma micro-silica (viz. KM, KM-6NS, and KM-6MS, respectively) and Fuji IX GP^®^ incorporating 0 and 6 wt% Aerosil^®^ OX 50 nano-silica or Aeroperl^®^ 300 Pharma micro-silica (viz. FIX, FIX-6NS, and FIX-6MS, respectively). Arrows indicate micro-silica particles.

**Figure 8 dentistry-12-00054-f008:**
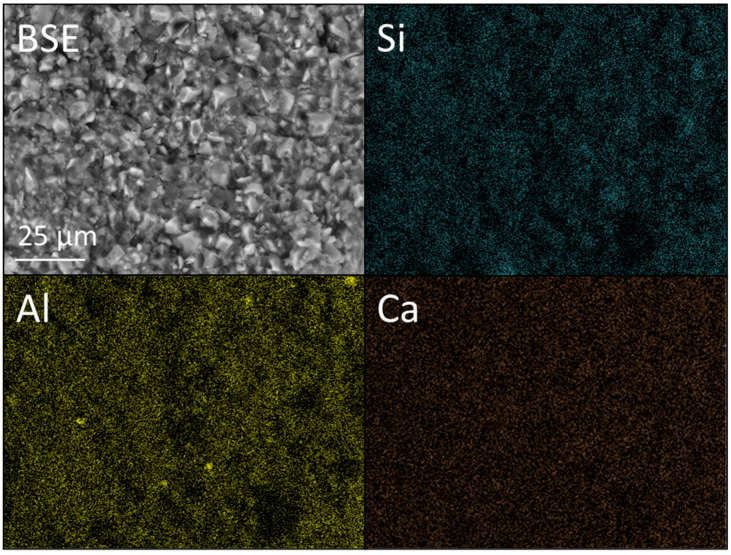
Backscattered electron (BSE) image and Si, Al, and Ca EDX maps of Ketac™ Molar (viz. KM).

**Figure 9 dentistry-12-00054-f009:**
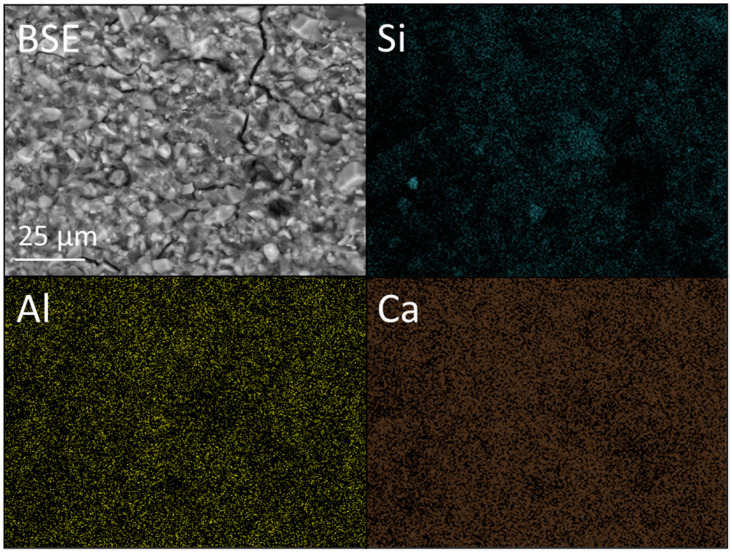
Backscattered electron (BSE) image and Si, Al, and Ca EDX maps of Ketac™ Molar incorporating 6 wt% Aerosil^®^ OX 50 nano-silica (viz. KM-6NS).

**Figure 10 dentistry-12-00054-f010:**
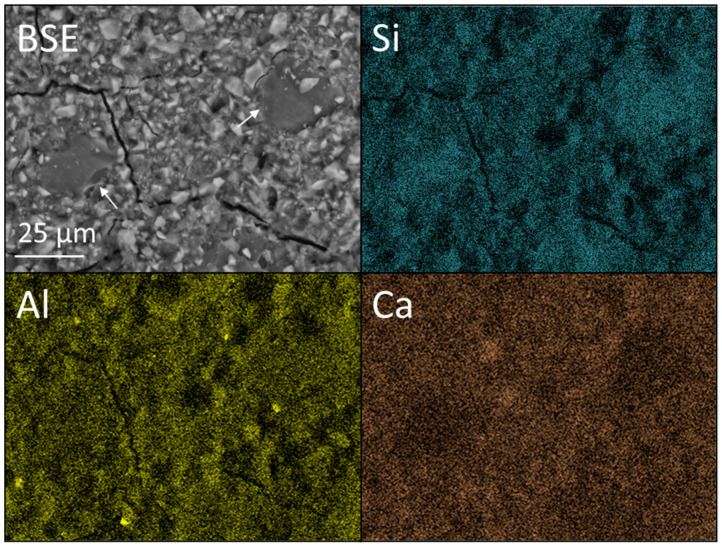
Backscattered electron (BSE) image and Si, Al, and Ca EDX maps of Ketac™ Molar incorporating 6 wt% Aeroperl^®^ 300 Pharma micro-silica (viz. KM-6MS). Arrows indicate micro-silica particles.

**Figure 11 dentistry-12-00054-f011:**
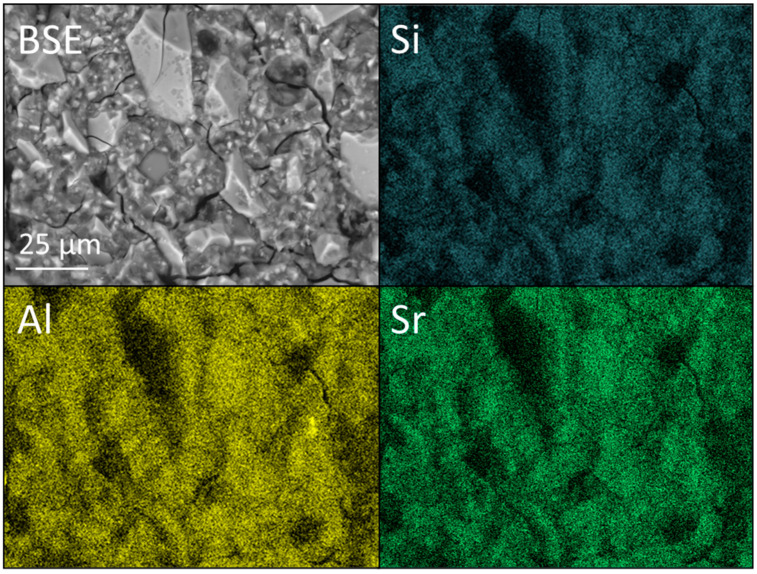
Backscattered electron (BSE) image and Si, Al, and Sr EDX maps of Fuji IX GP^®^ (viz. FIX).

**Figure 12 dentistry-12-00054-f012:**
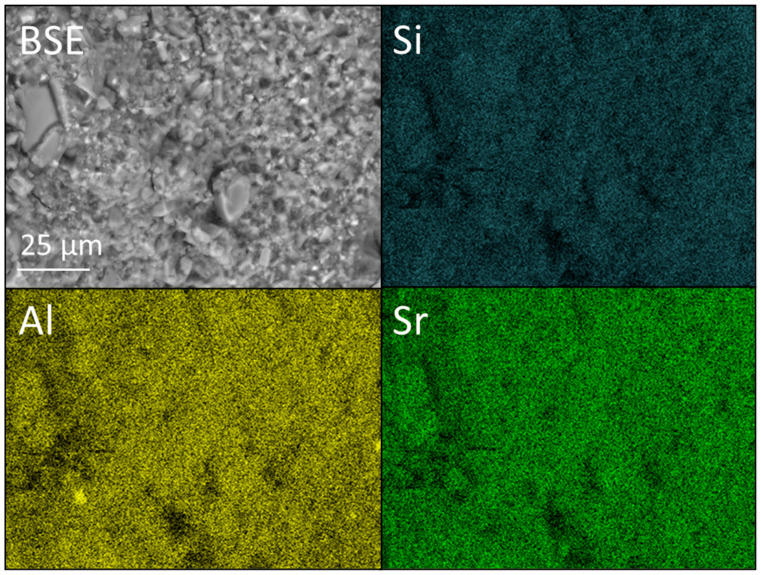
Backscattered electron (BSE) image and Si, Al, and Sr EDX maps of Fuji IX GP^®^ incorporating 6 wt% Aerosil^®^ OX 50 nano-silica (viz. FIX-6NS).

**Figure 13 dentistry-12-00054-f013:**
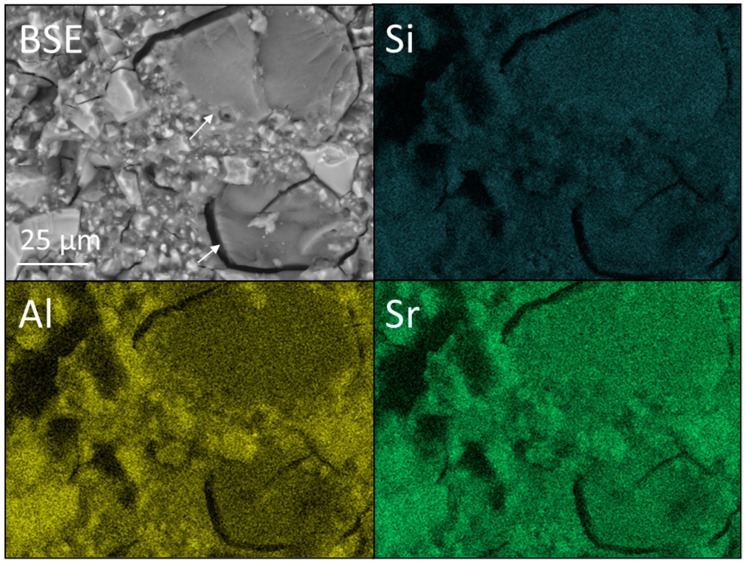
Backscattered electron (BSE) image and Si, Al, and Sr EDX maps of Fuji IX GP^®^ incorporating 6 wt% Aeroperl^®^ 300 Pharma micro-silica (viz. FIX-6MS). Arrows indicate micro-silica particles.

**Table 1 dentistry-12-00054-t001:** Composition and mix proportions of conventional commercial GICs [26,27].

GIC	Ketac™ Molar	Fuji IX GP^®^
Manufacturer	3M ESPE, St. Paul, MN, USA	GC Corporation, Tokyo, Japan
Constituents	Calcium fluoroaluminosilicate glass, poly(acrylic acid-co-maleic acid), tartaric acid and water	Strontium fluoroaluminosilicate glass, poly(acrylic acid), tartaric acid and water
Powder:liquid mass ratio	4.5:1.0	3.6:1.0

**Table 2 dentistry-12-00054-t002:** Properties of medical-grade nano- and micro-silica [28,29].

Silica	Aerosil^®^ OX 50	Aeroperl^®^ 300 Pharma
Manufacturer	Evonik Operations GmbH, Essen, Germany	Evonik Operations GmbH, Essen, Germany
SiO_2_ content	>99.8%	>99.0%
Phase content	X-ray amorphous	X-ray amorphous
pH	3.8–4.8	3.5–5.5
Mean particle size	40 nm	20–60 μm
BET surface area ^1^	35–65 m^2^ g^−1^	260–320 m^2^ g^−1^
Pore volume	-	1.5–1.9 cm^3^ g^−1^

^1^ Brunauer–Emmett–Teller surface area.

**Table 3 dentistry-12-00054-t003:** Mix proportions for the measurements of setting time and microhardness.

Sample Code	GIC Powder (g)	GIC Solution (g)	Aerosil^®^ OX 50 (g)	Aeroperl^®^ 300 Pharma (g)
KM	0.23	0.051	-	-
KM-2NS	0.23	0.051	0.0046	-
KM-4NS	0.23	0.051	0.0092	-
KM-6NS	0.23	0.051	0.0138	-
KM-2MS	0.23	0.051	-	0.0046
KM-4MS	0.23	0.051	-	0.0092
KM-6MS	0.23	0.051	-	0.0138
FIX	0.23	0.051	-	-
FIX-2NS	0.23	0.051	0.0046	-
FIX-4NS	0.23	0.051	0.0092	-
FIX-6NS	0.23	0.051	0.0138	-
FIX-2MS	0.23	0.051	-	0.0046
FIX-4MS	0.23	0.051	-	0.0092
FIX-6MS	0.23	0.051	-	0.0138

## Data Availability

Data are available on request from the corresponding author.

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
