# Peer review of "The Impact of Nano- and Micro-Silica on the Setting Time and Microhardness of Conventional Glass–Ionomer Cements"

_dentistry, 2024, doi:10.3390/dj12030054_

Round 1

Reviewer 1 Report

Comments and Suggestions for Authors

This study investigated changes of setting time and microhardness of two different GIC materials with different percentages of micro- and nano- silica particles. the research question is clear, and I have following suggestions to the authors.

1. methods part, please indicate the powder liquid ratio of GIC mixture. Since the GIC is manually mixed, more details on how to standardize the procedure is needed.

2. references are needed for setting time measurement.

3. I am curies since there is no ISO standard for hardness, why didn’t the authors use some existing article as reference. In my point of view, compressive strength is different from microhardness, microhardness has a high requirement of surface smoothness. More details of microhardness sample preparation are needed.

4. discussion part is kind of repeating the results and reviewing other’s findings, while no much discussion on the results, the possible reason behind the results. Improvement is needed.

Reviewer 2 Report

Comments and Suggestions for Authors

The authors investigated the effect of nano-micro-silica granules on the initial setting and early-stage microhardness of two types of clinically avaliable GIC products. It is very intersting the silica with different particle size level exhibited some different afficacy on physicochemical and mechanical properties. However, two questions should be clarified before publication:

1. Please provide some information (e.g. SEM images) involving the silica particle distribution in the GIC substrates because this is helpful to explain the different properties derived from the secondary phase. 

2. It is also valuable to provide the representative light images of the samples during the Vickers hardness testing process. 

Reviewer 3 Report

Comments and Suggestions for Authors

It is an interesting laboratory study in which authors have incorporated inorganic nano-particles to improve the properties of glass-ionomer cements. There are a few major concerns related to this study:

Abstract: 

1. What does this 2,4,6,% mean? is it weight % or volume %? Please clarify

2. Which statistical tests were used and what was the p value, please mention.

Introduction:

1. The introduction and the Discussion can be further improved. the auithors may use the following relevant paper:

Khan AA, Siddiqui AZ, Syed J, Elsharawy M, Alghamdi AM, Matinlinna JP. Effect of short E-glass fiber reinforcement on surface and mechanical properties of glass-ionomer cements. Journal of Molecular and Engineering Materials. 2017 Dec 21;5(04):1740007.

2. From lines 87-96, most of the text belongs to methodology section and the authors have already described in Table 2.The information related to nano and micro sized particles, manufacturers, shape, size and diameter needs to be removed.

3. What was the study's hypothesis and the answer of the hypothesis in the Discussion part is missing?

Methodology:

1. How much powder:liquid ratio was used for the control and experimental samples fabrication? please mention

2. What was the sample size per group?

3. Did the authors calculate the sample size prior to conducting the research?

4. Why the authors did not evaluate the flexural strength and tensile strength paramters?

5. Why the bonding properties of a new experimental GICs not evaluated?

6. Why the water solubility and sorption tests were not included in this research. These tests are very vital specifically for GIC material's testing

7. Why the film thickness test was not conducted?

Discussion:

1. What are the limitations and future recommendations for this study? Please mention 

2. What is the clinical implications of your findings? Write a short paragraph.

Round 2

Reviewer 1 Report

Comments and Suggestions for Authors

the suggestions have been addressed

Author Response

The reviewer indicates that we, the authors, have addressed all of their suggestions, so we have nothing further to add, other than to thank the reviewer for their time and attention. 

Reviewer 3 Report

Comments and Suggestions for Authors

The study has a serious flaw. A few of the important comments were ignored and were not addressed.

Author Response

As requested, we have now added information:

'Silica, one of the most abundant minerals in the Earth’s crust, is a popular constituent in many pharmaceutical formulations, foodstuffs, cosmetics, personal hygiene products and medical devices as it is simple and cheap to produce [53]. Furthermore, unlike many other metal oxide particles, silica exhibits low toxicity across a wide range of human cell-types and can stimulate hard tissue regeneration at a genetic level [56]. Hence, silica particles afford a number of clinical advantages over other INPs, and accordingly, further work is warranted to investigate the potential influence of nano- and micro-silica on the mechanical, chemical and biological properties of GICs. In this respect, further research on the impact of silica particles on the flexural, compressive and shear strengths, adhesive and bonding properties, solubility, fluoride-release and recharge, and biocompatibility of GICs would be relevant to an evaluation of their durability and potential clinical performance.'